# Preventing, Mitigating, and Treating Women’s Perinatal Mental Health Problems during the COVID-19 Pandemic: A Scoping Review of Reviews with a Qualitative Narrative Synthesis

**DOI:** 10.3390/bs13050358

**Published:** 2023-04-25

**Authors:** Pietro Grussu, Gianfranco J. Jorizzo, Fiona Alderdice, Rosa Maria Quatraro

**Affiliations:** 1Consultorio Familiare, South Padua District, Azienda ULSS 6 Euganea, Veneto Region, National Health Service, 35042 Este, Italy; 2Prenatal Medicine, Azienda ULSS 6 Euganea, Veneto Region, National Health Service, 35131 Padova, Italy; 3National Perinatal Epidemiology Unit, Nuffield Department of Population Health, University of Oxford, Oxford OX3 7LF, UK; 4School of Nursing and Midwifery, Queens University Belfast, Belfast BT9 7BL, UK; 5Maternità in Difficoltà^®^, 35138 Padova, Italy

**Keywords:** perinatal mental health, COVID-19, prevention, mitigation, treatment

## Abstract

Meeting the mental health needs of perinatal women during the COVID-19 pandemic is a serious concern. This scoping review looks at how to prevent, mitigate or treat the mental health problems faced by women during a pandemic, and lays out suggestions for further research. Interventions for women with pre-existing mental health problems or health problems that develop during the perinatal period are included. The literature in English published in 2020–2021 is explored. Hand searches were conducted in PubMed and PsychINFO using the terms COVID-19, perinatal mental health and review. A total of 13 systematic and scoping reviews and meta-analyses were included. This scoping review shows that every woman should be assessed for mental health issues at every stage of her pregnancy and postpartum, with particular attention to women with a history of mental health problems. In the COVID-19 era, efforts should be focused on reducing the magnitude of stress and a perceived sense of lack of control experienced by perinatal women. Helpful instructions for women with perinatal mental health problems include mindfulness, distress tolerance skills, relaxation exercises, and interpersonal relationship building skills. Further longitudinal multicenter cohort studies could help improve the current knowledge. Promoting perinatal resilience and fostering positive coping skills, mitigating perinatal mental health problems, screening all prenatal and postpartum women for affective disorders, and using telehealth services appear to be indispensable resources. In future, governments and research agencies will need to pay greater attention to the trade-offs of reducing the spread of the virus through lockdowns, physical distancing, and quarantine measures and developing policies to mitigate the mental health impact on perinatal women.

## 1. Introduction

Pregnancy and the postpartum are psychologically vulnerable periods in a woman’s life and the perinatal experience is accompanied by significant changes that occur in various physical, psychological, social, economic, and family dimensions. Although motherhood is generally considered to be a positive and unique experience, one in five pregnant or postpartum women has a diagnosed mood or anxiety disorder, which are the most common mental health conditions that occur during the perinatal period [1]. The prevalence of depressive episodes in the perinatal period ranges from 6.5% to 12.9%, with different values depending on the trimester of pregnancy or the period relating to the first year after delivery [2]. The incidence of more severe disorders is lower; for example, puerperal psychosis seems to range from 0.89 to 2.6 in 1000 women [3].

Psychological status and mental disorders in pregnant and postpartum women arise from a combination of effects, including sociodemographic factors, stress, and the availability of support from partners, families, societies, and nations. Specific sociodemographic aspects include age, parity, trimester, marital status, educational level, and socioeconomic status. Sources of stress include disaster or crisis, life events, marital dissatisfaction, and medical or obstetric complications [4].

Screening, assessment, diagnosis, referral to a mental health professional, treatment and follow-up are the main steps that characterize a structural intervention for the early identification of women at risk, their care and the safeguarding of their mental health in the perinatal period [5].

Early in 2020, Wuhan, China, was the site of an outbreak of an unfamiliar pneumonia characterized by atypical symptoms. The outbreak led to the identification and designation of the coronavirus disease 2019 (COVID-19). Faced with a significant challenge, the World Health Organization (WHO) monitored the rapid transmission of the disease from its occurrence in Wuhan to a major world pandemic. COVID-19 vaccines became a focal point for research, and by December 2020, more than 200 vaccines were being tested. Of those experimental vaccines, eight were ultimately approved by the FDA and EUA for use in humans [6]. In the absence of vaccines, mitigation was the key strategy in reducing the spread of the virus with a range of social restrictions that significantly impacted the care of women in the perinatal period and generated significant risk for women with mental health problems [7].

The COVID-19 pandemic added numerous risk factors for perinatal mental health [8]. Particularly important risk factors include the following: uncertainty about health risks (for example, the impact of COVID-19 on pregnancy outcomes); structural and organizational changes in maternity services; the inability of the partner to be present (or only partially present) during antenatal appointments in the labor room and/or in the delivery room; the reduced postpartum hospital stay of the mother–infant dyad; concerns about the management of COVID-19 patients within the hospital itself; reduced or delayed asking for help during pregnancy due to worries of contracting COVID-19 when attending appointments; fewer in-person prenatal appointments, thus reducing the frequency of checking the mother’s and baby’s wellbeing; and the separation of the newborns from COVID-19 positive mothers. Furthermore, the reduced availability of prenatal education classes for pregnant women and new mothers due to cancellations or delays could negatively affect women. In addition, the reduced frequency of medical visits and prenatal ultrasound scans can further reduce women’s opportunities to access timely and direct information from health professionals. In this way, the women are unable to reduce uncertainty and assuage their concerns about health risks. Fewer appointments during pregnancy can also reduce opportunities for healthcare professionals to heighten awareness of a range of health-promoting behaviors for mothers and babies, such as the positive benefits of breastfeeding and the importance of detecting any reduction in fetal movements in the third trimester [9].

With a view to reducing risk to both perinatal women and healthcare workers in the COVID-19 era, efforts have been made to limit face-to-face contact in healthcare settings as much as possible by increasing the use of online consultations. Prenatal visits using remote technologies—variously referred to as telehealth, mobile health, telemedicine, and eHealth—are designed to reduce pregnant women’s exposure to the virus while maintaining regular contact. Better attendance at virtual rather than in-person prenatal appointments and birth classes is accounted for anecdotally by the removal of barriers to attendance such as taking time off from work, arranging childcare, and paying for transportation [10]. Remote healthcare services have been significantly advanced through the use of video calls and other state-of-the-art telecommunication applications.

During the COVID-19 pandemic, pregnant and postpartum women constitute a population vulnerable to mental health disorders [11,12].

Recent meta-analyses have shown that in the course of the COVID-19 pandemic, symptoms of depression, anxiety, PTSD, stress, and sleep problems were prevalent not only during pregnancy but also after childbirth. Symptoms of depression were reported by 24.9% of women; 32.8% reported anxiety; 29.44% dealt with stress; 27.93% suffered PTSD, while 24.38% experienced sleep disorders [13]. Overall, the work of Delanerolle and colleagues [13] has revealed a distinct impact triggered by COVID-19 on the mental wellbeing of mothers during pregnancy and postpartum.

Meeting the mental health needs of these perinatal women is a priority [4]. Understanding the facilitators of and barriers to activities that promote good mental health is a strategic target that should lead to a call to action for mental health science when establishing multidisciplinary research priorities in relation to the COVID-19 pandemic and to inform future pandemics [14]. One particular area of mental health research should be an investigation into what can be done to prevent, mitigate or treat problems faced by vulnerable groups [15], such as pregnant and postpartum women, during pandemics.

Considering the peculiarities and inconveniences that COVID-19 has brought to light, identifying specific prevention methods can prevent perinatal mental health problems and their associated disabilities from presenting, developing, getting worse or coming back. Beyond the objectives of reducing the incidence, prevalence, and recurrence of mental health disorders, directing attention to the most recent considerations that have emerged within the COVID-19 literature can be of great help in daily clinical practice to identify ways to control stress and increase the resilience and self-esteem levels of perinatal women.

At the same time, identifying possible strategies to mitigate perinatal mental health problems—through processes and policies—can help configure improved measures to safeguard psychological wellbeing in pregnancy and after childbirth.

Finally, in order to build future treatment protocols, it may be of great use to collect and analyze new therapeutic actions or re-adapted treatment modalities suggested by perinatal mental health researchers in these first two years of the COVID-19 pandemic.

Using an explorative approach, we sought to conduct a scoping review: (a) On the prevention, mitigation, and treatment interventions of perinatal mental health disorders envisaged in a context of COVID-19; and (b) Of recommendations for research on public health interventions in regard to the pandemic.

In particular, the objective is to summarize the recommendations relative to preventing, mitigating or treating perinatal mental health problems reported in reviews regarding the COVID-19 pandemic published in 2020 and 2021. For completeness, we also summarize the suggestions proposed by the authors themselves regarding necessary further studies in the area of public health, which could add to our knowledge of preventing, mitigating or treating perinatal mental health problems.

## 2. Materials and Methods

### 2.1. Methodological Rationale

As little is known about the range of public health interventions used to prevent, mitigate or treat perinatal mental health problems during the pandemic, a scoping review with a qualitative narrative synthesis of findings was deemed to be more appropriate than a systematic review.

The purpose of this paper is to collect the descriptions of possible interventions reported in the literature to prevent, mitigate, and treat women’s perinatal mental health problems in the context of the current COVID-19 pandemic.

In addition, this manuscript is a collection of the reflections and final considerations reported by authors who have recently drawn together detailed reviews of clinical research on the link between the COVID-19 pandemic and women’s perinatal mental health.

As such, a scoping review was undertaken, and the five-stage process methodological framework outlined by Arksey and O’Malley [16,17] was followed. Additionally, the preferred reporting items for systematic reviews and the reporting guidelines for meta-analysis extension for scoping reviews (PRISMA-ScR) were used to direct the protocol followed.

### 2.2. Identify Research Question

The research question asked is as follows: what can be done to prevent, mitigate or treat women’s perinatal mental health problems during the COVID-19 pandemic?

### 2.3. Search Strategy

From the literature search, reviews published in English from 1 January 2020 to 31 December 2021 were included. Hand searches were conducted in EBSCOhost (PubMed and PsychINFO) using the terms: “COVID-19”, “perinatal mental health” and “review”. The terms were broad to be as inclusive as possible and in recognition that possible interventions reported in the literature to prevent, mitigate, and treat women’s perinatal mental health problems may be a part of a more general review of perinatal mental health and COVID-19.

### 2.4. Inclusion Criteria and Study Selection

Literature reviews were included if: (a) They were published during the time frame outlined above; and (b) They were primarily review studies of the direct or indirect effects of the COVID-19 pandemic on women’s perinatal mental health. Systematic and scoping reviews and meta-analyses were included. Reports, research, survey results, case reports, case series, observational studies, qualitative studies, qualitative analyses, and letters to the editor were excluded. This process yielded twenty-four peer-reviewed articles. Figure 1 details the PRISMA flow diagram of the study selection, providing more detail on this process.

### 2.5. Collating, Summarizing, and Reporting the Results

P.G. and R.M.Q. collated, summarized, and reported results with attention to the presence of elements concerning the: (1) Prevention; (2) Mitigation; (3) Treatment of women’s perinatal mental health problems in the COVID-19 pandemic context in each review; and (4) Suggestions for new studies in these areas.

After configuring the four sections and inserting the elements collected by P.G. and R.M.Q. in their analyses of the reviews, a consultation group (a trainee psychologist, a midwife, a senior social scientist, and a gynecologist) had the task of analyzing the consistency and appropriateness of what to report within these thematic sections. Additionally, the group cited further reviews in the literature and any articles that dealt with one or more items related to the topics considered in this manuscript.

Finally, the information collected was organized into the above-mentioned four sections, and a template was developed that includes: author(s) and date, aim of review, presence of elements concerning prevention, mitigation and treatment of perinatal mental problems, and suggestions for new studies in this regard (Table 1). A qualitative narrative of the findings was then produced to provide an overview of the key findings.

## 3. Results

The preliminary search of electronic databases yielded twenty-four articles. Eight did not reach the inclusion criteria—they were clinical studies, notes, and comments—and one paper was a duplicate. In all, thirteen reviews were examined for intervention for the prevention, mitigation or treatment of women’s perinatal mental health problems in the context of the COVID-19 pandemic, as well as proposals for further studies in these areas. Four thematic sections were expanded and are reported in full below. Table 1 provides an overview of the authors and articles that have provided contributions to each section.

### 3.1. Section 1: Preventive Primary Interventions Aimed at Perinatal Mental Health Problems in the Context of the COVID-19 Pandemic

The authors of nine reviews commented on primary prevention actions that could be activated in the COVID-19 era, proposing multiple interventions, as follows.

Planning for healthcare service delivery during the current pandemic must take into account the particular need for dependable prenatal care by vulnerable individuals and groups [18].

Sharing information about COVID-19 should begin as early as possible. There are multiple reliable methods to disseminate this information, including the websites of governments, health departments, and health facilities. Additionally, there are toll-free telephone lines that are interactive or operated by knowledgeable staff. Hardcopy or digital materials can be distributed during in-person or online office visits or when arranging appointments for those visits [12].

Healthcare professionals can be involved in mental health promotions [12], and interventions for perinatal mental health should also be a priority [20] for women with healthy pregnancies [19].

The prevention of mood and anxiety disorders before and after childbirth can be critically influenced by perinatal planning. Psychoeducation during pregnancy enables a woman to anticipate and solve problems and to garner support before childbirth. That same preparation prevents her from relapsing after delivery [26]. In general, assisting a woman as she evaluates the risks and benefits of each decision is a significant service [26].

Clinicians are in an ideal position to deliver psychoeducation regarding lifestyle modifications and to help women create their own personalized perinatal strategy for engaging in activities that prevent or minimize the strain of mood or anxiety symptoms. An effective plan will recognize ways to mobilize social supports, overcome barriers to self-care, and lead to maximizing sleep, nutrition, and exercise. Support for the mother can be rallied by including family members and other caregivers in the discussion. While the pandemic has created barriers to the implementation of lifestyle modifications, it has also introduced opportunities [26].

Ideally, the discussion about postpartum support should begin before childbirth, and should include dialogue on self-care and the prevention of perinatal mood and anxiety disorders. Even if mothers have not had these conversations before delivery, hospital clinicians can still play an integral role in providing psychoeducation after birth. During a public health emergency, such as the pandemic, women have minimal support from family or friends in hospital during labor and delivery. At such times, the assessment and support provided by hospital clinicians becomes indispensable. Postpartum nurses are well-positioned to broaden their instructions to include vital information that includes mental health care, as well as physical health care. A woman’s plan for her postpartum support network should be in place and confirmed by clinicians prior to discharge from hospital [26].

During the COVID-19 pandemic, it is more important than ever to carry out activities to promote awareness among perinatal women, identify symptoms of depression and anxiety [12], and to help women evaluate the risks and benefits of each choice as they plan for postpartum support prior to delivery [26].

All women should be assessed for mental health issues at every stage of their pregnancy and postpartum [12]. An accurate screening approach should be implemented [8] because screening can help identify women who warrant further mental health evaluation and support [26]. This is particularly applicable at times when the healthcare system is unable to respond to an escalation in the demand for services. Such is the situation arising out of the emergency created by the COVID-19 pandemic. The burden on the healthcare system can be reduced when screening first identifies the most vulnerable women and they are targeted for intervention [8].

Healthcare providers should guarantee easy access to mental health services as a primary strategy to prevent long-term impacts on individuals and to support the health of both mothers and children [20]. It is important to continue to monitor the mood of postpartum patients, especially women with anxiety and depressive disorders in pregnancy [12]. Prenatal visits can be considered as an opportunity to assess women who are at risk, and ensure the necessary support is put in place to help them during this particularly vulnerable time [23].

Prior to the postpartum visit, and even while still in hospital after delivery, many women, particularly those at higher risk for perinatal mood and anxiety disorders and those reporting symptoms, would benefit from screening. After identifying these higher-risk women using a validated instrument and clinical assessment, clinicians could connect them to resources before their discharge from hospital or follow them more closely during their postpartum period. These measures take into account the insufficiency of support from a woman’s usual sources as a result of the pandemic [26]. Standardized tools and questionnaires can also be used remotely. Telehealth options may be used in screening and monitoring perinatal mental health [12].

However, the reduction of hospital stays after childbirth and the conversion of the postpartum control visit to the telematic mode should not be allowed in order to diminish the possibility of early detection of women at risk or already suffering from perinatal mental disorders [22].

The mental status of pregnant and postpartum women should receive extra attention by healthcare workers, especially by physicians, during routine appointments [20]. The clinicians’ investigations should address perceived stress, symptoms of mental health problems, depression/anxiety symptoms, available social support, and psychosocial vulnerabilities [12]. When conducted in perinatal healthcare settings, the assessment should take suicide risk into consideration [12]. Globally, in the tradition of person-centered care, each family’s risk analysis should differ, taking into account the family’s individual needs, resources, and risks, and the woman’s vulnerability to postpartum psychiatric illness [26].

Obstetric and mental health appointments throughout the prenatal and postpartum periods are essential, even during the COVID-19 epidemic, and it is important to help a woman to assess the risks and benefits of attending those appointments in person or virtually [26]. Certain groups of patients are at higher risk of depression or anxiety during the COVID-19 health emergency and particular attention should be paid to them. Those higher-risk groups include single women without social support, younger women, women in financial difficulty, and the unemployed. Additionally, included are women who experience high stress of becoming infected, those who have a high-risk pregnancy or chronic illness, or those with a previous psychiatric diagnosis and previous adverse occurrences during pregnancy [12].

Public health messaging must emphasize the importance of prenatal care and provide avenues of support for those at risk of intimate-partner violence [18].

With social distancing being the new normal, the relevance of social support is amplified. In circumstances where the public health system is under stress, pregnant women in low-resource settings are supported by non-governmental organizations, which play a vital role through integrated community-based activities, focused on reducing household food insecurity and improving social support to advance maternal mental health [25].

### 3.2. Section 2: Mitigation Interventions Aimed at Perinatal Mental Health Problems in the Context of the COVID-19 Pandemic

The authors of seven reviews commented on the mitigation interventions that could be activated in the COVID-19 era, proposing multiple actions, as follows.

Perinatal mood and anxiety disorders can be mitigated during the pandemic by making the best use of a mother’s contact with her current clinicians. The clinical team might include general mental health clinicians, family practitioners, obstetrician-gynecologists, and pediatricians [26].

It is evident that policies requiring a lockdown must be countered by other measures designed to mitigate the impacts of those policies [24].

During the pandemic, tasks such as navigating social systems, modifying social relationships, and incorporating exercise into routines may become more problematic. In such cases, clinicians can provide practical problem-solving guidance that may help attenuate perinatal mood and anxiety disorders [26].

The COVID-19 pandemic highlights the need for timely and individualized interventions to ease mental problems among pregnant and postpartum women. This especially applies to multigravida women and women in the first and third trimesters of pregnancy [4]. Regardless, understanding a woman’s risks for relapse enables intervention to mitigate those risks [26].

At home, during the COVID-19 emergency, pregnant women can practice self-care to improve their mental health and wellbeing through mind–body interventions, such as yoga, mindfulness, and relaxation exercises [25].

Perinatal women should be encouraged to focus on certain specific aims: reducing their level of stress and perceived sense of lack of control; increasing their capacity to cope and increasing the level of social support; and promoting adequate sleep and exercise. Taking into account the evidence that the COVID-19 distress of a perinatal woman and that of her partner are linked; preventative measures should involve both parents [10].

The vital importance of social contact remains unchanged during the pandemic, while opportunities for establishing such connections have shifted. Beyond the social aspect, perinatal women also benefit from hands-on help. That practical assistance ranges from direct infant care to household chores, and allows perinatal women to focus on self-care. It is important to broaden a woman’s search for diverse sources of support, including childcare. Although face-to-face meetings may be curtailed, technology enables pregnant and new mothers to form social connections virtually. It is important to circulate information about these social gateways among perinatal women who may not be aware of them. Virtual contact can be of special importance to women in rural, geographically isolated areas, as well as to women with childcare needs that prevent them from connecting in person with other perinatal women. Thanks to virtual platforms, extended families in distant locations can reduce perinatal women’s feelings of isolation. Beyond the constraints of the health emergency, new and expecting mothers can meet in person outdoors, observing pandemic guidelines for masking and social distancing. Whether in person or online, support networks and the introduction of partner sessions are essential to mitigate the risk of perinatal mood and anxiety disorders [26].

Finding time for regular exercise in the postpartum period becomes even more of a challenge for new mothers with limited support because of the COVID-19 pandemic [26]. Healthcare professionals can intervene by collaborating with pregnant women to plan physical activities and encouraging them to participate in social opportunities [12]. Women who engage in regular physical exercise despite the shutdown of indoor and outdoor recreational spaces during the pandemic may improve their maternal mental health, especially when compared to sedentary women. Online fitness sessions appear to be a practical alternative. When there are no contraindications, physical activity for mothers should always be encouraged [20]. Additional activities such as gardening, going for walks while observing physical distancing guidelines, carrying out household chores, and practicing yoga at home are feasible options to improve depression outcomes [25].

New barriers mounted by the COVID-19 pandemic make it even more difficult to receive help for babies’ nighttime care, which would maximize mothers’ sleep during postpartum. It is important to guide a woman in her assessment of the COVID-19 risks of introducing a new caregiver into the home or of exposing vulnerable elderly grandparents to the family, and to consider strategies to mitigate potential COVID-19 exposure. Following maternal COVID-19 vaccination, the evaluation of breastfeeding or pumping compared to giving formula overnight entails new risks and benefits, because the breast milk now contains SARS-CoV-2-specific antibodies [26].

It is crucial to distinguish whether a postpartum woman’s decreased sleep is due to caring for the infant, or whether there is an inability to sleep even when the infant is sleeping. Interventions in the two situations will differ, and the latter instance may involve more troubling psychiatric symptomatology [26].

During the pandemic, connecting online can be a safe and easy option for patient support [20]. With the increase in virtual opportunities, incorporating online classes into patient support programs is becoming easier [26].

Resilience factors can be intentionally pursued as part of mental health prevention strategies at individual and system levels. Catalysts for resilience include: virtual communication platforms; self-care elements such as adequate sleep, physical activity and healthy diet; partner emotional support; and spending time outdoors [12]. Involving family members in discussions can help protect mothers’ self-care time [26].

### 3.3. Section 3: Treatments Targeting Perinatal Mental Health Problems in the Context of the COVID-19 Pandemic

The authors of three reviews commented on the treatments that could be activated in the COVID-19 era, proposing multiple interventions, as follows.

Collaboration between providers of perinatal and mental health care services is essential for women with perinatal mental health problems. Recognizing the need for consultation and treatment by mental health professionals is critical [12]. Intentionally pursuing resilience factors is a legitimate element of mental health intervention strategies at individual and system levels. When access to psychotherapy is impeded, obstacles must be addressed [26].

A woman who experiences increased distress during the COVID-19 pandemic can benefit from empirically based interventions that assess her capacity for control and that result in a realistic appraisal of potential risks while, at the same time, working towards reducing stress and enhancing coping skills. Exercises focused on mindfulness, distress tolerance, relaxation, and interpersonal relationship skills can be of assistance to this population [10]. Short-term interventions targeting improved self-efficacy and motivating participation in healthy activities could be helpful in decreasing anxiety [12].

Restrictions that were put in place during the COVID-19 pandemic compelled the prenatal care system to redirect efforts towards new virtual visits. The value of telehealth in psychiatric care in prenatal cases has been confirmed in the COVID-19 situation. Virtual visits make it possible to quickly reach people in need, permitting screening, monitoring, and therapy of mental disorders while observing social distancing restrictions [12].

Psychiatric consultation and psychological support should be offered to women with mental health issues, and urgent psychiatric hospitalization should be initiated if necessary. Treatment will vary, depending on the condition of the patient, and the risk of suicide must always be considered. Therapy for patients requiring support and counselling during the pandemic often takes place online. Based on the most recent scientific evidence, telehealth options may be used to treat perinatal mental health disorders detected during the COVID-19 pandemic era [12].

On the other hand, once in-person contact is safe again, the use of digital communication and information technologies should not entirely replace face-to-face treatment for patients, especially those needing intensive mental health treatment and support. Access to digital platforms might not be available to all pregnant women in low-resource settings, especially those in the lower quintiles of socioeconomic strata. The digital divide created by this inequitable access to care will be further broadened if face-to-face services are widely forsaken in favor of digital platforms. Socially vulnerable women may suffer the most [25].

### 3.4. Section 4: Future Studies on Perinatal Mental Health Problems in the Context of the COVID-19 Pandemic

The authors of six reviews recommended future studies to increase knowledge on how to prevent, mitigate, and treat women’s perinatal mental health problems in the COVID-19 context, proposing the following.

Longitudinal multicenter cohort studies should be conducted to reinforce standardized screening and intervention guidelines in support of pregnant and postpartum women and to promote healthy family behaviors during the COVID-19 pandemic [8]. Prioritizing concurrent healthcare needs will require high-quality research on which to base responses to potential future waves of COVID-19 and/or outbreaks of other infectious diseases [24]. Considering the mental health consequences of a widespread outbreak of infectious disease, health professionals must develop better support and reassurance tools to cope with the impact of the illness [20].

There is an urgent need for innovative cross-sectional and longitudinal cohort studies that include more diverse samples, observational assessments, and a control group [9,19]. It may be of further use to consider research designs that refer to a previous cohort as a comparison group in order to understand contextual changes and their impact on maternal/child wellbeing. Identifying which mitigation efforts are having a direct effect on population prevalence levels is another area requiring further research [9].

Funding bodies are eager to work with researchers and mothers with lived experience of postnatal depression [9]. Increased numbers of participants need to be included in future studies to determine the significance of variations on scales measuring depression and anxiety [20].

It is critical to take a closer look at factors that may increase the risk of postnatal depression in a pandemic context generally, and in the COVID-19 setting specifically [9]. Bearing in mind the long-term effect that maternal mental health has on child development, it is particularly important to identify the risks and protective factors of the current pandemic and to follow up women and their families to explore the longer-term impact [8]. Social determinants, such as female gender, low education, lower household income, and low level of social capital, have to be evaluated and ranked, when applicable, as risk factors for adverse mental health outcomes and related health inequities in pregnant women [25]. Expanding the currently limited knowledge base on the subject of maternal postnatal depression risk following crises, particularly pandemics, could enhance future decision-making by governments and healthcare providers [9].

Further intervention studies focused on screening and managing perinatal mental health through virtual means are needed [20]. Originally triggered by the distress linked to the COVID-19 pandemic, the pioneering of online methods to detect psychological problems and to deliver early mental health interventions for mothers and their infants will be of ongoing value [8].

Interventions that have previously been shown to be effective in ameliorating maternal mental health must be adapted to the current situation, and their effectiveness in the “new normal” and in new crises must be assessed. There is a further need for efforts to integrate evidence-based mental health interventions into routine maternal health care. Ultimately, policies and guidelines circulated by health ministries and other agencies need to pay due attention to improving the physical and mental health and wellbeing of pregnant and postnatal women [25].

## 4. Discussion

To assess the consequences of the COVID-19 crisis, public health researchers and healthcare professionals have initiated numerous studies and have engaged in considerable debate. Thirteen reviews of these studies have been identified in this scoping review. Topics incorporated into health policies have included the detection of distress in the general population and in health workers, the re-analyses of priority interventions in primary care, hospitals and clinics, and the maximization of healthcare resources. Due to the strict social restrictions widely imposed globally, attention to safeguarding perinatal mental health immediately became a widespread priority.

When information and health knowledge about the consequences of the COVID-19 pandemic were still scarce and fragmented, Thapa and colleagues [27] pointed out in a special editorial that during times of crisis, it is vital to watch for emerging threats to pregnant women and infants. The immediate healthcare issues in managing the impact of infection continue to overshadow mental health considerations. Although sufficient reliable evidence remains to be developed, at this time it can be reasonably speculated that pregnant women are at increased risk of developing mental health problems, such as depression, anxiety, and post-traumatic stress symptoms. In 2020, women in the UK were three times more likely to die by suicide during or up to six weeks after the end of pregnancy compared to 2017–2019 [28]. Taking into consideration that mental health is a leading cause of postpartum death in many countries, we need to renew our efforts to support maternal mental health. We also need longitudinal cohorts to identify the length of time and severity of symptoms and the ongoing significance of the problem for women and their families.

The multiple reviews on the COVID-19 pandemic and perinatal mental health published to date in major scientific journals confirm the presence of increased levels of distress [29] presenting as anxiety, depression, stress disorders, and sleep disturbance in pregnant and postpartum women [13].

This scoping review specifically explored the most recent reviews on COVID-19 and perinatal mental health with recommendations regarding prevention, mitigation, treatment, and future studies.

### 4.1. Primary Prevention

Information about COVID-19 should be provided as early as possible by healthcare workers, and healthcare professionals should be involved in the promotion of mental health [12]. Before childbirth, clinicians are in the ideal position to provide psychoeducation on lifestyle modifications, and to help women create their individualized perinatal plan that identifies activities that help prevent or lessen the impact of mood or anxiety symptoms [26]. It should be compulsory for every perinatal woman to be assessed for mental health issues at every stage of her pregnancy and postpartum [12]. Prenatal and postpartum visits should be considered an opportunity to assess women who are at risk, and to ensure that the necessary support is put in place to help them during this particularly vulnerable time. The administration of a validated instrument for screening together with a clinical assessment would benefit many women [23,26]. Suicide risk should always be included in the mental status assessment. Standardized tools and questionnaires can also be used remotely, especially in this pandemic context. Patients at risk of depression or anxiety warrant particular attention. These include women who are single and without social support, who are younger, who are facing financial difficulties, who are unemployed, who are at high risk of infection, as well as women with high-risk pregnancies, with chronic illness, with previous psychiatric diagnosis, and with previous adverse experiences during pregnancy [12].

### 4.2. Mitigation

Efforts should focus on reducing the magnitude of perinatal mothers’ stress and perceived sense of lack of control, while simultaneously expanding their capacity to cope and their level of social support, and promoting sufficient sleep and exercise [21]. By providing practical guidance in solving problems, clinicians may help abate perinatal mood and anxiety disorders [26]. Viable options for improving depression outcomes include physical activities, such as gardening, walking while maintaining physical distance, housework, yoga at home, and online exercise sessions [25]. Using virtual platforms to connect with family and friends who live at a distance can reduce the perinatal woman’s sense of isolation [26]. Those same virtual platforms can also serve as a safe and easy opportunity for patient support [20]. As access to virtual options continues to increase, it may become easier to organize online classes [26].

### 4.3. Treatment

When perinatal women experience mental health problems, collaboration is essential between clinicians providing perinatal care and those providing mental health services. It is crucial to recognize when a perinatal woman needs to be referred to a mental health professional [12]. Further consideration could be given to the development of formalized collaborative models to aid continuity of care for women offering good support and clear pathways to specialized mental health care. In addition, perinatal women can benefit from practicing mindfulness, doing relaxation exercises, and refining their skills for tolerating distress and enhancing interpersonal relationships [21]. Treatment must also take into account the risk of suicide as an element of the patient’s condition. Some patients, for example, may need only psychological support and counselling. In general, digital communication technologies can be used in the treatment of perinatal mental health patients [12], but in-person treatment should not be abandoned in favor of virtual visits for patients in extreme need and for those who require intensive mental health treatment [25].

### 4.4. Future Studies

There is a particular need for timely cross-sectional [9] longitudinal multicenter cohort studies that can advance our knowledge on standardized screening and intervention guidelines for supporting pregnant and postpartum women if another wave of the COVID-19 pandemic occurs [8]. These studies should include more diverse samples, observational assessments, and a control group [19]. Taking into consideration the long-term effect of maternal mental health on a child’s development, the identification of social determinants [25], and risks and protective factors arising out of the pandemic is of particular importance [11]. Furthermore, to demonstrate the significance of the variations on the depression and anxiety scales, the number of women recruited to participate in future studies should be increased [20], and they should be followed up in the year after birth to explore the duration and severity of symptoms. The development of innovative online methods of detecting psychological problems and delivering early mental health interventions to mothers and their infants will be valuable [8]. Evidence-based mental health interventions should also be integrated into routine maternal health care [25].

### 4.5. Considerations

Most of the suggestions on prevention, mitigation, and treatment reported above are elements already widely present in the literature prior to the COVID-19 pandemic [30] as necessary and expected interventions of good clinical practice to safeguard perinatal mental wellbeing. In particular, the limitations related to reducing the spread of the virus necessarily led to the widespread use of telehealth care visits, telehealth classes, and e-screening for perinatal mental health. Most often, these methods have been used in an experimental way in response to the disruptive health emergency. In the future, the field of perinatal telehealth in particular needs multiple verifications of efficiency, effectiveness, and appropriateness.

### 4.6. Limitations

One major limitation of this scoping review was linked to the choice to extract interventions on prevention, mitigation, and treatment present in the reviews and not in the individual studies of perinatal mental health problems in the COVID-19 era. While our approach provides a valuable overview of the current literature, innovative intervention may have been missed in reports of single studies.

Finally, the fact that the individual suggestions on prevention, mitigation, treatment, and new studies were developed by researchers at different stages in the spread of the virus should not be overlooked. For this reason, about two years after the first COVID-19 infections, some indications are outdated or partial.

## 5. Conclusions

Perinatal mental health disorders are wide-ranging, and can arise for the first time during the perinatal period or may represent a relapse of a pre-existing condition. These disorders include depression, anxiety disorders, and postpartum psychosis, which usually manifests as bipolar disorder.

While the full impact of the COVID-19 pandemic is not yet understood, there is general acknowledgement that perinatal distress and restrictions to perinatal mental health services have materialized worldwide [27]. Although there has been some amelioration of risks as a result of policy and clinical responses, this health emergency has amplified the occurrence of perinatal mood disorders. An increase in the number of women needing support for perinatal mental health problems has been reported everywhere.

In the COVID-19 era, online programs and the delivery of virtual health services facilitate opportunities to provide holistic care to women and their partners as they progress towards improved mental wellbeing [31]. Such programs may be of benefit moving forward.

Fostering perinatal resilience and positive coping, mitigating perinatal mental health problems, screening all prenatal and postpartum women for depression/anxiety, and using telehealth services seem to be further confirmed as indispensable resources for meeting health needs in the perinatal period during and following the COVID-19 pandemic.

Creative approaches to solving pandemic-induced problems have led to the innovative use of technology in healthcare settings as multidisciplinary teams provide original and effective assistance to patients and their families [31].

Perinatal individuals and their infants have clearly been negatively affected by the COVID-19 pandemic. Knowing this, the academic community, healthcare providers, and policy makers now have an obligation to learn from experience. Humanitarian disasters often have an adverse effect on women’s healthcare [32,33]. The pressing need to plan maternity services that will carry on through any emergency is emphasized in numerous studies reported in the literature [18].

In the future, governments and research agencies will need to pay greater attention to the trade-offs of reducing the spread of the virus through lockdowns, physical distancing, and quarantine measures and developing policies to mitigate the impacts for those who are extremely vulnerable, such as perinatal women. In this regard, some studies suggest that the increased rate of adverse outcomes might result primarily from the inefficiency of healthcare systems and their failure to cope with the pandemic, rather than from the stringency of pandemic mitigation measures [18].

The pandemic should be seen as an opportunity to forge better, stronger, more resilient societies, capable of bringing relief and hope to women worldwide [34].

## Figures and Tables

**Figure 1 behavsci-13-00358-f001:**
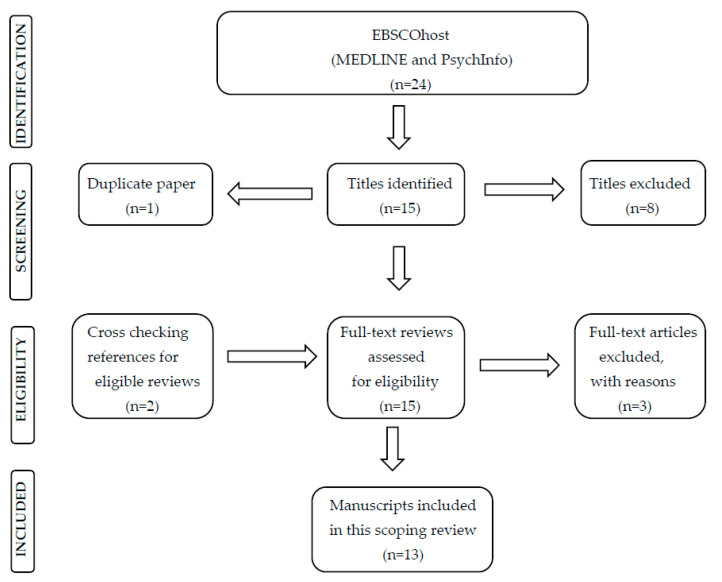
PRISMA flow chart documenting steps of the literature search.

**Table 1 behavsci-13-00358-t001:** Papers considered in this scoping review and presence of elements concerning prevention, mitigation, and treatment of perinatal mental problems, and suggestions for new studies.

Authors,Date	Declared Objectives	Perinatal Mental Health:Presence of Interventions on
Prevention	Mitigation	Treatment	Recommended Future Studies
Ahmad and Vismara, 2021[8]	“This review aimed to investigate the existing literature on the psychological impact of the COVID-19 pandemic on women during pregnancy and the first year postpartum”.	✓			✓
Chmielewska et al., 2020[18]	“We aimed to assess the collateral effects on maternal, fetal, and neonatal outcomes of the global COVID-19 pandemic”.	✓			
Doyle and Klein, 2020[9]	“The aims of this paper were therefore to conduct a review to identify the established risk factors for maternal postnatal depression and generate evidence-based hypotheses about whether the COVID-19 crisis would likely increase or decrease postnatal depression rates based on the identified risk factors”.				✓
Feduniw et al., 2021[19]	“This study aimed to investigate the impact of anxiety on the mental health of pregnant women exposed to catastrophic events as compared to those without such exposure”.	✓			✓
Hessami et al., 2020[20]	“The present systematic review and meta-analysis aimed to assess the influence of the COVID-19 pandemic and subsequent physical distancing/isolation measures on women’s mental health during pregnancy and perinatal period”	✓	✓		✓
Iyengar et al., 2021[21]	“…we review perinatal mental health outcomes in the context of key obstetric and neonatal outcomes that have been documented to date”.		✓		
Kotlar et al., 2021[22]	“The goal of this scoping review is to synthesize the current literature on both the direct consequences of contracting COVID-19 during pregnancy and the indirect consequences of the pandemic for pregnant individuals and mothers…”	✓			
Ryan et al., 2020[23]	“The aim of this article is to review the current data in relation to how COVID-19 affects pregnant women, the information to date on treatment options, its psychological impact and its wider effect on healthcare services and resources”.	✓			
Russo et al., 2021[24]	“The aim of this research was to synthetise the existing evidence on the impact of epidemic-related lockdown measures on women and children’s health in low- and lower-middle-income countries (LLMICs)”.		✓		✓
Shidhaye et al., 2020[25]	“…we provide a high-level overview of the recently published epidemiological studies assessing the impact of COVID-19 on maternal mental health, and propose an integrated approach to improve mental health and well-being of pregnant women during the current crisis”.	✓	✓	✓	✓
Susser et al., 2021[26]	“…we present a perinatal planning guide with practical tips for mitigating risk of perinatal mood and anxiety disorders…both during and outside of the COVID-19 pandemic”.	✓	✓	✓	
Suwalska et al., 2021[12]	“The aim of the current study was to analyze the prevalence and intensity of mental health problems in pregnant women and new mothers during the COVID-19 pandemic, examine risk factors of depression and anxiety, as well as protective factors, and consider the application of the findings in perinatal care”.	✓	✓	✓	
Yan et al., 2020[4]	“The aims of this systematic review and meta-analysis are to quantify the influence of the COVID-19 pandemic on the mental health of pregnant and postpartum women, and to explore the specific vulnerable groups among this population of women”.		✓

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
