# Peer review of "Preventing, Mitigating, and Treating Women’s Perinatal Mental Health Problems during the COVID-19 Pandemic: A Scoping Review of Reviews with a Qualitative Narrative Synthesis"

_behavsci, 2023, doi:10.3390/bs13050358_

Round 1

Reviewer 1 Report

The authors addressed a major concern but require a more thorough literature review.

Reviewer 2 Report

Dear Authors. Very interesting review on perinatal mental health. However, you have to pay attention to some points.

 In general, the introduction suffers from a lack of references. 

Motherhood is not always seen as a positive experience. Revised

"Obstetric women". Do you mean "postpartum women" or "pregnant women"?

"directing attention to the most recent suggestions..." who are they? 

The purpose of your study is inconsistent with the title

why were so few databases used?

you are not showing the search terms. Your search returned very few articles. It would be useful to see the search terms

what do you mean "future studies" in the table 1?

Why is the type of prevention not analyzed?

Reviewer 3 Report

This is a very interesting paper reviewing the prevention and treatment of perinatal mental health problems in women during the COVID-19 pandemic. The paper is really of interest for the journal; however, several minor changes should be made before considering it for publication.

ABSTRACT

1- At the beginning of the introduction, the authors should clarify if they are reviewing the occurrence of mental health problems (primary prevention and treatment) or the appearance of recurrences or relapses in women previously diagnosed. 

INTRODUCTION

1- I recommend to expand the introduction section when focusing on basic concepts on the motherhood. More references are needed.

MATERIAL AND METHODS

1- Screening and selection processed should be better clarified.

2- What does "consultation exercise" mean?I consider that this term is confusing. 

RESULTS

1- Preventive interventions are prefered to Prevention interventions. This section should clarify those refering to primary or secondary prevention.

In the conclusions, the authors recommend preventive strategies in the perinatal period. I suggest to divide the findings into primary and secondary prevention. 

The concept of perinatal mental health problems are really extensive. I recommend to define which kind of problems are they referring. Anxiety and depressive symptoms, or mental health disorders. 

The conclusions should be considered a separate section (for instance, section 5).

Round 2

Reviewer 1 Report

The manuscript has been revised appropriately.

Author Response

Thanks for your positive feedback.

Your previous comments and those of other reviewers have allowed us to significantly improve the description of the Scoping review.

For the quality of the English language, a further careful reading of the manuscript was carried out.

Reviewer 2 Report

Dear Authors

From what I saw you made a great effort to correct the article. The major problems have been eliminated

Author Response

(The authors gave the same response as above.)
